# Anion Exchange Membranes for Alkaline Polymer Electrolyte Fuel Cells—A Concise Review

**DOI:** 10.3390/ma15165601

**Published:** 2022-08-15

**Authors:** Hari Gopi Kuppusamy, Prabhakaran Dhanasekaran, Niluroutu Nagaraju, Maniprakundil Neeshma, Baskaran Mohan Dass, Vishal M. Dhavale, Sreekuttan M. Unni, Santoshkumar D. Bhat

**Affiliations:** CSIR-Central Electrochemical Research Institute-Madras Unit, CSIR Madras Complex, Chennai 600113, India

**Keywords:** anion exchange membrane, polymer membrane, durability, fuel cell, synthesis route

## Abstract

Solid anion exchange membrane (AEM) electrolytes are an essential commodity considering their importance as separators in alkaline polymer electrolyte fuel cells (APEFC). Mechanical and thermal stability are distinguished by polymer matrix characteristics, whereas anion exchange capacity, transport number, and conductivities are governed by the anionic group. The physico-chemical stability is regulated mostly by the polymer matrix and, to a lesser extent, the cationic head framework. The quaternary ammonium (QA), phosphonium, guanidinium, benzimidazolium, pyrrolidinium, and spirocyclic cation-based AEMs are widely studied in the literature. In addition, ion solvating blends, hybrids, and interpenetrating networks still hold prominence in terms of membrane stability. To realize and enhance the performance of an alkaline polymer electrolyte fuel cell (APEFC), it is also necessary to understand the transport processes for the hydroxyl (OH^−^) ion in anion exchange membranes. In the present review, the radiation grafting of the monomer and chemical modification to introduce cationic charges/moiety are emphasized. In follow-up, the recent advances in the synthesis of anion exchange membranes from poly(phenylene oxide) via chloromethylation and quaternization, and from aliphatic polymers such as poly(vinyl alcohol) and chitosan via direct quaternization are highlighted. Overall, this review concisely provides an in-depth analysis of recent advances in anion exchange membrane (AEM) and its viability in APEFC.

## 1. Introduction

Due to the obvious availability of well-established infrastructure across the world, fossil fuels continue to lead the worldwide energy market. However, decreasing supplies, skyrocketing prices, and their impact have urged for a rapid shift to clean and renewable energy sources. Fuel cells have grown in popularity for their excellent efficiency and low pollution. In fuel cells, electrochemical processes are employed to convert the chemical energy of fuels into electrical energy. Electrical efficiency in numerous fuel cell units are frequently better than in classic thermal power plants [1,2,3,4,5]. In this regard, alkaline fuel cells (AFCs) are among the different types of fuel cells with aqueous potassium hydroxide as a liquid electrolyte and have the highest efficiency. The National Aeronautics and Space Administration (NASA) adopted AFC technology in space missions (Apollo program) in the 1950s, and the KOH liquid electrolyte played a significant role in OH^−^ ion conduction. In an AFC, OH^−^ ion is produced and transported from the cathode side to the anode side. Despite its early success, fuel cell technology is not well advanced due to material issues, and other challenges associated with it [1,2,3,4,5,6,7,8,9]. In electrode reaction of AFC, water is generated at the anode [10].
2H_2_ + 4OH^−^ → 4H_2_O + 4e^−^(1)
O_2_ + 2H_2_O + 4e^−^ → 4OH^−^(2)
2H_2_ + O_2_ → 2H_2_O(3)

The use of the KOH liquid electrolyte solution, which is particularly sensitive to CO_2_, may combine with the hydroxyl (OH^−^) ions to generate K_2_CO_3_ and is one of the key disadvantages of AFC.
CO_2_ + 2KOH → K_2_CO_3_ + H_2_O(4)

Due to this, OH^−^ concentration may decrease and will affect the overall anion conductivity in AFC. In addition, carbonate salt formation on the electrodes may block the pores of gas diffusion layer, thus reducing the reaction kinetics of the fuel cell. Another disadvantage of utilizing liquid electrolytes is that if the liquid is less or more, it can cause drying or electrode flooding and there is a possibility of leakage of electrolytes if not sealed properly. As a result, the necessity for high-purity gas with low CO_2_ content in the oxidant supply stream, CO_2_ sensitivity, and electrolyte leakage associated with the use of liquid KOH electrolyte were key limitations [10]. To overcome these bottlenecks and complications, it is possible to utilize an anion exchange membrane (AEM) as a solid polymer electrolyte membrane instead of liquid KOH electrolyte.

Alkaline polymer electrolyte membrane fuel cell (APEFC) technology has seen a revival due to the growing use of solid electrolytes and non-precious metal electrocatalysts for the electrode reaction. This resulted in prominent breakthroughs in the literature on polymer electrolytes for AFCs. Many reports on AFC innovation have been published over the years, with most of the reports appearing in the past five years [11]. In principle, in APEFCs, OH^−^ ion is transported from the cathode side to the anode side via an alkaline electrolyte membrane, and due to their alkaline cell environment, they have numerous potential benefits in comparison to polymer electrolyte fuel cell (PEFC) technology [12]. In this respect, the major advantages are the use of precious group metal (PGM)-free catalysts for oxygen reduction reaction (ORR) kinetics, enhanced durability of materials, and flexibility to use non-fluorinated electrolyte membranes. In addition, in case of APEFC, when methanol is used as a fuel, fuel crossover may decrease because the path of hydroxide ion transport is contrary to the path of methanol crossover.

AEMs are solid polymer electrolytes with positively charged ionic carriers (typically phosphonium or quaternary ammonium type) covalently bonded to the polymeric matrix. By adding a movable counter-ion towards each ionic substituent, polymer electro-neutrality is retained [12,13]. The chemical stability of AEM is regulated by the polymer framework and by the cationic moieties. To realize and improve cell performance, it is essential to understand the possible transport mechanism of OH^−^ ion in an AEM. OH^−^ transport takes place via Grotthuss’ mechanism, diffusion, migration, and convection, as represented in Figure 1. OH^−^ transport exhibits Grotthuss characteristics in aqueous medium [14,15,16]. According to this approach, the anionic species diffuses across the hydrogen-bonded framework of the chain of water molecules via covalent bond construction or dissociation. The diffusive transport arises only in the presence of any electric potential gradient or concentration. Convective transport occurs when OH^−^ ions move across the barrier, dragging water molecules with them, resulting in a convective flow of water molecules within the barrier [17].

Since water in the system functions as a constant dipole and interacts with the fixed charges of the membrane, surface site hopping of hydroxyl anion transportation across the membrane is regarded to be of secondary type. Tuckerman et al. proposed that the transport of OH^−^ ion is facilitated by protons passing through the hydrogen-bonded matrix of water molecules via development and breaking of O-H bonds endorsed by molecular reorganization and rotation of water molecules using the Grotthuss mechanism [18].

The weak basicity of quaternary cation sites having pK_b_ in the range of 4 accounts for the inferior mobility of OH^−^ in relation to H^+^ in dilute solutions, i.e., ion mobility of H^+^ = 4.76, whereas that of OH^−^ is 2.69, which could constrain the ionic conductivity of AEMs [19]. When compared to PEMs, AEMs have a higher water self-diffusion coefficient but weaker water binding, resulting in low ionic conductivity and pressure-driven water permeability. According to morphology, AEMs are transformed to the ionic form during surface modification, whereas sulfonated polymers are formed directly in their ionic state. Among those stated above, conductivity and alkaline stability are the most significant characteristics of an AEM. There are several ionic head group compositions currently available that operate as charge transfer for OH^−^ ionic conduction and increase membrane physical and chemical properties [12,13]. The cationic group is more crucial to achieve a high concentration of charges in the membrane while maintaining adequate ionic conductivity, and it will also have a direct impact on chemical stability. Thus, AEMs are synthesized with high ion-exchange capacity and (or) with microphase structure separation to improve the OH^−^ conductivity. Specifically, the chemical properties of both the polymeric chains and the ionic moieties influence an AEM’s alkaline stability. The strong nucleophilicity and functional group of hydroxide ions may cause the breakdown of the polymer backbone and covalently attached cation sites [20]. As an outcome, the number of anionic exchange sites will decrease, resulting in decreased ionic conductivity. The cationic head groups that have been synthesized and investigated in the literature include quaternary ammonium [21,22,23,24,25], imidazolium [26,27,28], guanidinium [29,30], phosphonium [31], tertiary sulfonium [32], spirocyclic quaternary ammonium [33], quaternary phosphazenium [34], benzimidazolium [35], and pyrrolidinium [36]. The alkaline stability of these organic cations is affected by various factors such as conjugative effect, steric hindrance effect, electron donation effect, and σ-π hyper-conjugative effect of the substitutes on cation [20].

Earlier, much APEFC research focused on the synthesis of an effective anion exchange membrane (AEM), but with only few studies providing real cell activity and long-term stability studies. A stable synthesis pathway with optimal selection and alignment of cationic ion-exchange groups, membrane architecture to enhance mechanical characteristics, flexibility, and utilization of low-cost production processes is of paramount importance for the development of an AEM. AEMs are classified into three types based on their architecture and synthesis method: (i) interpenetrating polymer networks, (ii) heterogeneous membranes, and (iii) homogeneous membranes [12,13]. In the present review, we explain in detail the importance of cation groups in the preparation of an AEM and its performance evaluation. In addition, several polymer composite membranes realized in APEFCs and their structure, performance, and limitations are discussed in the present review.

## 2. Cation-Based AEMs

### 2.1. Quaternary Ammonium (QA) Cation Based AEMs

Quaternary ammonium (QA) groups provide comparatively high OH^−^ conductivity, adequate alkaline stability throughout time scale, and simplicity of surface modification. However, QA cations are often unstable at high temperatures owing to Hofmann elimination (E2), nucleophilic substitution (SN2), and (or) ylides synthesis (Y) [37]. To overcome this issue, recent reports suggest that the decomposition of the AEMs is dependent on the aliphatic chain length of amines; hence, the longer the alkyl chain length, the lower the degradation of AEMs. QA groups that are attached or copolymerized with long alkyl N-bound sequences may have strong alkaline durability.

Leng et al. designed and synthesized quaternized poly(2,6-dimethyl phenylene oxide)s with extended alkyl side chains attached to the same nitrogen-centered cation, as represented in Figure 2a. The conductivities of comb-shaped membrane with one alkyl group chain outperformed those of benzyltrimethyl ammonium cations [38]. The steric effects of the long alkyl chains surrounding the QA group may improve the alkaline stability. In the same vein, Jannasch et al. introduced a series of poly(phenylene oxide) (PPO)-based AEMs by attaching QA classes to PPO directly in benzylic position or via heptyl spacer and flexible pentyl units [39]. The results showed that the alkyl spacers improved the alkaline stability of AEMs in 1 M NaOH at 80 °C. In addition, phase separation morphologies with hydrophilic zones will be produced by introducing pendant alkyl side chains. Li et al. developed a comb-shaped anion exchange membrane with nanoscale morphology, resulting in enhanced hydroxide conductivity [40]. Cationic groups incorporated on pendant alkyl filler sequences can greatly improve hydroxide OH^−^ conductivity and stability. Gopi et al. developed varieties of AEMs with variable degrees of quaternization of polyvinyl alcohol (PVA) with alkyl spacers, as represented in Figure 2b [41]. Before being tested for ionic conductivity and ion exchange capacity, these as-synthesized materials were thermally and chemically cross-linked. According to the X-ray diffraction analysis, the inclusion of a quaternary ammonium framework breaks the crystal lattice of the polymer structure, enlarging the amorphous phase, hence improving ionic conductivity. At 40 °C, the quaternized poly(vinyl alcohol) (QPVA) with 15 wt % alkyl fillers shows excellent conductivity of 0.0057 S cm^−1^ compared to the other ratios. At 40 °C, cell polarization data for the optimized QPVA membrane with varied ionomer concentrations were also investigated. In addition, the QPVA-based alkaline membrane shows excellent stability even after 25 h. Since alkyl spacer hexadecyl trimethyl bromide (HDT) comprises an alkyl spacer (-CH_2_-) with a long chain placed between the polymeric matrix, resulting in the available cationic groups, this may improve anion transport properties, thereby enhancing membrane conductivity.

Understanding the transport mechanism for hydroxyl ions, as well as significant efforts to boost conductivity while preserving durability, are required before AEMs can be produced and utilized in fuel cell technology. Unlike the traditional approach of benzyl substitution, chloromethylation of polymer on the aryl carbon is less sensitive to crowding. The effect of chemical modification by altering the amine to polymer ratio was noteworthy since effective quaternization not only boosts electrochemical properties but also membrane longevity. Gopi et al. reported a poly(phenylene oxide) (PPO)-based membrane used for APEFCs [42,43]. As depicted in Figure 3, facile chloromethylation of PPO was accomplished through aryl and benzyl substitution, followed by homogeneous quaternization. The impact of several factors on the chloromethylation process, such as the volume, polymer type, and reaction temperature, were explored. PPO-based membranes in APEFCs delivered a peak power density of 111 mW cm^−2^. The increasing cell performance and stability of quaternary ammonium functional groups are attributed to aryl chloromethylation. To further elucidate the impact of metal concentrations in the catalyst layer, the electrocatalyst loading is varied and optimized based on alkaline direct methanol fuel cell (ADMFC) performance employing QPPO 3 membrane substrate. MEAs containing the variable loading condition of Pt-Ru/C and Pt/C on both the anode and cathode, such as 2, 1, and 0.5 mg cm^−2^, were investigated and are illustrated in Figure 3c. The metal loading of the catalyst was raised from 0.5 mg to 2 mg cm^−2^ to improve cell performance. Generally, AEM degradation is often feasible under basic circumstances owing to the decomposition of either the cationic head group or the polymer backbone. The loss of stable QA ions diminishes the ion exchange capacity, and hence decreases conductivity, whereas a breakdown of the polymer structure causes brittleness in AEMs. The alkaline stability of the modified QPPO 3 membrane was tested by immersing the membrane sample in a 4 mol dm^−3^ KOH solution for 300 h at 75 °C. QPPO 3 membrane shows excellent stability even after 100 h at OCV condition while using 2 M methanol as a fuel, as shown in Figure 3d [42].

### 2.2. Phosphonium Cation Based AEMs

Several research groups have also investigated quaternary phosphonium-based AEMs in the APEFC. According to Bauer et al. the phosphonium cation was unfavorable for AEMs due to its quick decomposition in 1 M NaOH solution [44]. According to Ramani et al., trimethylphosphonium cations have significantly lower alkaline longevity than trimethylammonium cations [45]. A possible reason is that the phosphonium cations degrade rapidly through the ylide intermediate, as phosphorous is more polarizable when compared with nitrogen. In contrast, Yan et al. developed stable AEM with a tris(2,4,6-trimethoxyphenyl) phosphonium cation, showing good alkaline stability [31], as the trimethoxyphenyl groups protect the phosphorus atom against the hydroxide ion attack. The charge carrier characteristic of oxygen in the phosphonium moiety preserves the cation, resulting in high chemical stability even under basic environment conditions. Coates et al. synthesized polyethylene-based AEM using tetrakis-(dialkylamino) phosphonium cation as a charge carrier, illustrating no degradation even after immersing in 15 M KOH solution at 22 °C and 1 M KOH at 80 °C for 140 and 25 days, respectively, represented in Figure 4 [46].

### 2.3. Guanidinium Cation Based AEMs

Guanidinium cation has a positive charged center delocalized over carbon and between three nitrogen atoms and shows increased ionic conductivity compared to the trimethylammonium-functionalized polymer because of its higher basicity. Only a few methods are available in the literature to prepare guanidinium cations, which include fluorophenyl-amine reaction and nucleophilic substitution polymerization. Zhan et al. reported guanidinium-grafted poly(aryl ether sulfone) membranes via nucleophilic substitution polymerization, succeeded by the interaction of diamine groups with Vilsmeier salt, as represented in Figure 5 [29]. Li et al. fabricated phenyl-, benzyl-, and methyl-substituted guanidinium cations attached to polymer matrix, of which the methyl-substituted guanidinium cation has the better stability due to the higher electron-donating effect of the methyl group [47].

### 2.4. Benzimidazolium Cation Based AEMs

Poly(benzimidazole) is known for its chemical and thermal stability because of the strong interaction between imine nitrogen of the benzimidazole chain and the amine proton. Xu et al. prepared benzimidazolium (BIm)-functionalized AEMs from PPO backbone via the attachment of methylbenzimidazole groups [35]. The nucleophilic substitution process of BPPO-X with MBIm formed BIm-PPO, and the resulting polymer was designated as BIm-PPO-X, as shown in Figure 6a. Similarly, the related OH^−^ conductivity of the fully hydrated QPPO, BIm-PPO, and Im-PPO is compared with function of IEC, as shown in Figure 6b. The optimum level of BIm-PPO has enhanced IEC value, which may be due to BIm cation interaction, which plays an important role in the improvement of ionic conduction.

BIm-PPO-0.54, which will have the best fuel cell activity, such as the greatest ionic conductivity and IEC and better mechanical characteristics, was tested in aqueous KOH for 7 days at 25 °C, and the values of WU, IEC, and σ during the operation were measured and the comparable results are shown in Figure 6c. The optimum level of BIm-PPO-0.54 nearly reduced the IEC value of 18%, WU of 27%, and σ of 35%, respectively. This may be due to the density of BIm cations in the membrane. The similar trends observed in polybenzimi dazolium-based AEM membranes may be ascribed to the spontaneous ring-opening of BIm cations, which may be attacked by hydroxyl ions. The optimum level of the BIm-PPO-0.54-based membrane shows excellent fuel cell performance and delivers maximum power density of 13 mW cm^−2^ at a cell temperature of 35 °C, as represented in Figure 6d.

The synthesized AEMs reveal increased ionic conductivity and improved mechanical and thermal stabilities compared to those of analogous PPO AEMs containing pendant QA and imidazolium cations. Recently, Henkensmeier et al. explored that the hydroxide formation of benzimidazolium may show mesomeric stabilization. This enhances the stability of the benzimidazolium cation due to steric crowding of adjacent bulky groups around the reactive C2 position. This implies that shielding the C2 position of the benzimidazolium ring is an effective method to hinder the nucleophilic hydroxide ion attack [48].

### 2.5. Pyrrolidinium and Spirocyclic Quaternary Ammonium Cation Based AEMs

Considering their non-aromatic composition, pyrrolidinium-based cations show higher chemical stability compared to conventional QA. Yan et al. prepared pyrrolidinium cations with several N-substitutes (including methyl, ethyl, butyl, octyl, isopropyl, 2-hydroxylethyl, benzyl, and cyclohexylmethyl groups). Among them, the N, N-ethylmethyl-substituted pyrrolidinium was more stable than imidazolium and benzyltrimethyl ammonium cations [49]. Marino et al. observed that spirocyclic QAs are durable even at higher pH of 10 M NaOH [37]. Jannasch et al. designed and synthesized bis-N-spirocyclic quaternary ammonium (QA) AEM by introduction of spirocyclic QA classes in aromatic poly(arylene ether sulfone) backbone, as shown in Figure 7a [33]. To assure full conversion, QA groups were incorporated into polymers using Menshutkin reactions containing benzyl or alkyl halide groups in the polymer framework and an excess of a tertiary amine, e.g., trimethylamine. However, the synthesized AEMs deteriorated at high temperatures in alkaline medium, most likely owing to ring-opening replacement at the benzylic site. The conductivity of OH- and Br- in 100% hydrated AEM was studied with respect to temperature via electrochemical impedance spectroscopy, as represented in Figure 7b. This result clearly indicates that both OH- and Br- conductivity enhanced with respect to temperature and increasing IEC. While increasing the temperature above 40 °C, OH- conductivity of PAES-spiro-pip, PAES-spiro-pyr, and PAES-spiro-aze reached 31, 52, and 19 mS cm^−1^, respectively.

Figure 7c displays the water uptake characteristics of the aforementioned AEMs in both the OH^−^ and Br^−^ forms. The water uptake properties clearly indicate that they only slightly increased with respect to temperature. As predicted, after alteration into OH^−^ form, the water uptake of all three AEMs increased sharply. Similarly, Sata et al. reported that polysulfone and fluorocarbon-type polymers have demonstrated greater resistance to backbone decomposition, and various AEMs have been constructed via radiation-induced grafting. Regarding this, degradation of the polymer backbone generally results from the hydroxyl attack on polymer directly, which has an impact on the mechanical property and conductivity of membranes [50].

## 3. AEMs with Ion-Solvating Blends

The ion-solvating salt complex is a water-soluble polymer blended with hydroxide salt that acts as the matrix. A donor–acceptor linkage connects the polymer comprising electronegative heteroatoms such as nitrogen, oxygen, or sulfur to the salt cation. The above heteroatom–cation interactions influence the flexibility of amorphous molecular chains, in turn providing ionic conduction inside the structure. Poly(ethylene oxide) (PEO) was the most commonly used polymer because of its tendency to align with metal ionic species and efficiently solvate cations by interacting with polar ether networks to form homogenous polymer films [51]. Fauvarque et al. created PEO- and KOH-based ion-solvating blends [52,53,54,55,56]. Equivalent quantities of PEO and KOH were dissolved in methanol and physically agitated at room temperature for several hours and allowed the solvent to completely evaporate. The final residues of solvents were extracted using vacuum after casting on an adequate support. At ambient temperature, the ionic conductivity of the polymer increased up to 10^−3^ S cm^−1^. However, when salt concentration increases, ionic conductivity decreases, wherein PEO tends to crystallize. An alkaline electrolyte was produced by combining PVA with poly(epichlorohydrin) (PECH) [57]. When PECH was added to the PVA polymer network, the amorphous area expanded and an anion conductivity of 2 × 10^−2^ S cm^−1^ was obtained at ambient temperature. Yang et al. also synthesized the combination of tetraethyl ammonium chloride (TEAC) and PVA at room temperature. The above ionic membrane shows excellent anionic transport number (0.82–0.99) and ionic conductivity (2 × 10^−2^ S cm^−1^), as represented in Figure 8 [58].

AEMs were synthesized by blending PVA with acrylic acid monomer, followed by free radical polymerization [59]. The KOH/PAA/PVA membrane shows excellent ionic conductivity of around 30 × 10^−2^ S cm^−1^. There has been much effort put into developing various types of ion-solvating electrolytes exhibiting conductivities of 10^−2^ S cm^−1^. Consequently, owing to the prevalence of KOH in the framework, the stability of this membrane is quite low. Especially for APEFC applications, AEMs were mainly constructed/designed with PVA surface modification with quaternary ammonium groups.

## 4. Hybrid and Interpenetrating Polymer Network Based AEMs

Hybrid membranes are a class of membranes combining both an organic moiety that provides electrochemical characteristics and an inorganic part providing the mechanical properties. Generally, the sol–gel method is commonly used to create organic–inorganic hybrid membranes [60]. Wu et al. designed composite membranes from (N-triethoxysilylpropyl-N,N,N-trimethylammonium iodine) with trimethoxy silyl functionalized in a sol–gel technique, obtaining a PEO-[Si(OCH_3_)_3_]_2_ membrane [61]. Sang and Yang et al. also synthesized a composite membrane using titanium dioxide (TiO_2_) fillers in PVA matrix for APFC applications [62,63]. Herein, titania nanostructures act as a solid plasticizer which enhances the membrane electrochemical characteristics. Yang et al. proposed the fabrication of a nanocomposite PVA/ZrO_2_ membrane by directly incorporating ZrO_2_ fillers with particle sizes from 20 to 30 nm with PVA and KOH solution. This particular hybrid PVA-ZrO_2_ electrolyte with different ratios shows outstanding thermal stability and excellent ionic conductivity (267 × 10^−3^ S cm^−1^ at 20 °C), as represented in Figure 9a [64]. Kumar et al. reported a cross-linked poly(vinyl alcohol)-poly(acrylonitrile)-co-2-dimethylamino ethylmethacrylate-based AEM membrane, as shown in Figure 9b. A PVA-PAN-co-2-DMAEMA-based membrane was prepared via three different steps, i.e., the sol–gel process followed by chemical cross-linking and quaternization [65]. During the reaction with trimethylamine, the epoxy groups of the methacrylate copolymers altered the quaternary ammonium framework. Despite being mechanically stronger, the conductivity of these organic/inorganic filler-based membranes is lower than that of their homogeneous equivalents.

A cross-linked polymer matrix is made up of two different polymers, one of which is produced or interpenetrating in the availability of another, but there is no covalent bond between them. Cross-linked polymer matrix is made up of a hydrophobic polymer that is chemically, physically, and thermally stable, and AEMs based on cross-linked PVA-poly(acrylonitrile-co-2-dimethylaminoethylmethacrylate) were studied [66]. PVA was chosen for its strong chemical reactivity, which is advantageous for cross-linking, and poly(acrylonitrile-co-2-dimethylaminoethylmethacrylate) copolymer for improving chemical, mechanical, and thermal characteristics. Lebrun et al. developed semi-interpenetrating network (sIPN) thin AEMs by dissolving poly(vinyl imidazolium bromide) in PVA and cross-linking it using 1,2-dibromoethane, showing an IEC of 1.1 meq g^−1^ [67]. The ionic conductivity of various concentrations of PDVIBr-based membrane shows higher conducting nature with different types of electrolytes, as represented in Figure 9c. Altmeier et al. developed an IPN from poly(acrilonitrile) and epichlorohydrin, which was then quaternized and cross-linked using 4,4-diazabicyclo-[2.2.2]-octane (DABCO). The obtained conductivity was 2 × 10^−3^ S cm^−1^ with acceptable mechanical characteristics [68,69]. Shahi et al. produced a cross-linked network from poly(styrene-codivinylbenzene) copolymer and polyethylene (PE) that was chloromethylated and quaternized, and the membranes delivered IECs of up to 0.78 meq g^−1^ [70].

**Figure 9 materials-15-05601-f009:**
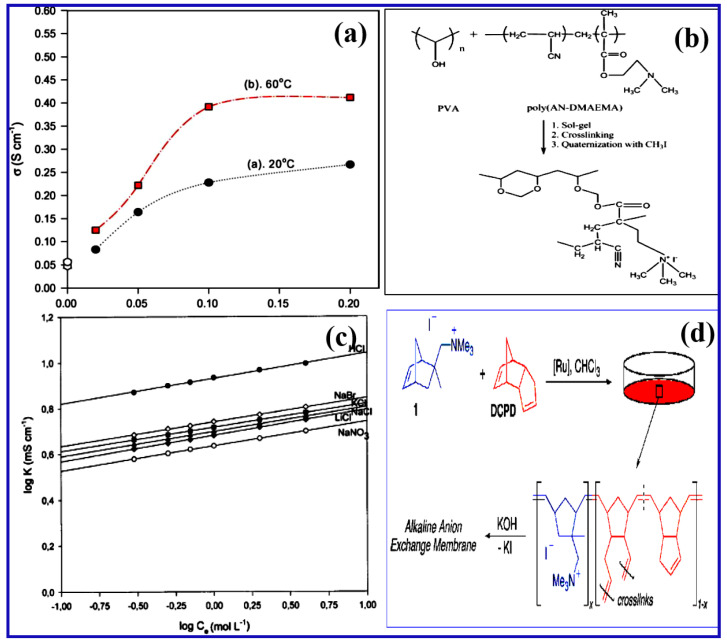
(**a**) The anionic conductivity values with respect to concentration of ZrO_2_ fillers in PVA membrane at two different temperatures of 20 and 60 °C. (Reproduced with permission from Ref. [64] Copyright © 2006.) (**b**) Formation of PVA-PAN-co-2-DMAEMA-based membrane with three different steps via sol–gel, cross-linking, and quaternization. NaBr. (Reproduced with permission from Ref. [65] Copyright © 2010.) (**c**) Bi-logarithmic variations of ionic conductivity with respect to concentration for PDVIBr-based membrane and for various electrolytes such as HCl, NaNO_3_, LiCl, KCl, NaCl, and NaBr. (Reproduced with permission from Ref. [67] Copyright © 2004.) (**d**) Synthesis of alkaline exchange membrane (AEM) by copolymerized tetraalklylammonium-functionalized norbornene with dicyclopentadiene. (Reproduced with permission from Ref. [71] Copyright © 2009).

Clark et al. copolymerized a tetraalklylammonium-functionalized norbornene with dicyclopentadiene by ring-opening metathesis employing a Grubbs second-generation catalyst (Figure 9d), resulting in conductivities ranging from 1.3–1.9 × 10^−2^ S cm^−1^ at ambient temperature to 2.0–2.7 × 10^−2^ S cm^−1^ at 50 °C [71]. Interpenetrating polymer networks are found to be inferior to heterogeneous membranes in terms of ion exchange capacity and ionic conductivity. Table 1 summarizes various parameters for the polymers subjected for AEM.

## 5. Homogenous AEM Membranes

In homogeneous electrolyte membranes, the ionic charges such as quaternary ammonium linkage are covalently cross-linked to the polymer network. To maintain the polymer electro-neutrality, a mobile counter ion is bonded to each ionic functional group. The stability of this type of electrolyte is affected by both temperature and alkaline media. The AEM shows exceptional stability for both the polymer backbone and the fixed cationic charges. There are several methods available to prepare these types of homogenous membranes.

AEMs are synthesized directly from polymer films by grafting a functionalized monomer or by monomer modification. Various techniques are usually used to graft copolymerize monomers onto polymer films. To irradiate a polymer, one of two sources of radiation is used: a particle beam (electrons) or electromagnetic radiation (X rays). Mostly, cobalt (^60^Co) or cesium (^137^Cs) isotopes are used as γ-ray sources. Varcoe and Slade et al. introduced the radiation-induced graft polymerization of chloromethyl styrene (CMS) onto fluorinated poly(tetrafluoroethylene-co-hexafluoropropylene) (FEP) and ethylene-co-tetrafluoroethylene (ETFE), as shown in Figure 10a. The film is irradiated with an electron beam and then immersed in vinyl benzyl chloride solution [72]. The resulting grafted poly(vinylbenzyl chloride) copolymer is soaked in trimethylamine solution to obtain quaternary ammonium groups. FEP-based AEMs observed better conductivities in the range of 1–2 × 10^−2^ S cm^−1^ at room temperature [73]. QA-functionalized radiation-grafted ETFE alkaline AEMs (Figure 10b) exhibit enhanced conductivity of 34 × 10^−3^ S cm^−1^ at 50 °C with an IEC of 0.92 meq g^−1^ [74]. In addition, the I-V characteristics of the AEMs displayed power density of 55 mW cm^−2^ for the AEM of thickness of ~153 μm.

Matsuoka et al. prepared 4-vinylpyridine-based AEMs via the plasma polymerization process [75]. Membranes derived from this process show ionic conductivity of 0.53 × 10^−3^ S cm^−1^ and low resistance. Hwang et al. explored amplified electron radiation to cross-link a polysulfone, leading to improved coulombic efficiency. Poly(epichlorhydrine) (PEC) exhibits high gas impermeability and chemical stability [76]. Since aliphatic polyethers are stable in alkaline medium, Agel et al. used polyether-based epichlorohydrin to prepare AEMs with 1,4-Diazabicyclo[2.2.2]octane (DABCO) and by further mixing it with trimethylamine [77]. Stoica et al. synthesized AEMs made of poly(epichlorohydrin) copolymer, cross-linked with allyl glycidyl ether [78]. To introduce anionic moiety, two cyclic diamines DABCO and 1-azabicyclo-[2.2.2]-octane (quinuclidine) were incorporated. After thermal cross-linking, the membrane showed better conductivity of 1.3 × 10^−2^ S cm^−1^ at 60 °C with an ion exchange capacity of 1.3 meq. g^−1^.

Choi et al. synthesized AEMs from PVA and 4-formyl-1-methylpyridinium benzene sulfonate for electrodialysis application [79]. Even though the formed PVA-FP membrane showed lower electrical resistance, it is not suitable for fuel cells because of the chemical instability of pyridinium groups in alkaline medium. Xiong et al. grafted quaternary ammonium group charge carriers onto the PVA backbone by using (2,3-epoxypropyl) trimethylammonium chloride and further cross-linked with glutaraldehyde [80]. The conductivities of cross-linked quaternized PVA (QPVA) membranes showed 2 to 7 × 10^−3^ S cm^−1^ conductivity at ambient temperature. Fu et al. reported an alkaline-doped PVA membrane by subjecting poly(ethylene glycol) dimethyl ether (PEDGE) and poly(vinyl pyrrolidine) (PVP) to the polymer network, which functions as a plasticizer and improves ionic conductivity and chemical stability [81]. Similarly, by a combined chemical and thermal cross-linking approach, an AEM was synthesized from PVA/poly(diallyl dimethylammonium chloride) (PDDA)-OH^−^ [82]. PVA was the polymer in the study, while PDDA was used as the anion charge carrier.

Chitosan derivatives such as N-[(2-Hydroxy-3-trimethylammonium) propyl] chitosan chloride were prepared and cross-linked with glutaraldehyde. Wan et al. synthesized a series of quaternized chitosan derivatives (QCDs) with IECs and conductivities ranging from 0.33 to 0.86 meq. g^−1^ and 4.8 mS cm^−1^ to around 8.0 mS cm^−1^, respectively, with glycidyltrimethyl ammonium chloride [83]. Pan et al. proposed a simple and cost-effective synthesis process for producing quaternary ammonium polysulfone. At room temperature, the membrane had an ionic conductivity of 10^−2^ S cm^−1^ and high mechanical strength [84]. Park et al. synthesized chloromethylated polyarylether sulfone (PSF)-based AEMs with three distinct amination methods: diamine, monoamine, and their combination [85]. Polysulfone membranes were also cross-linked through epoxy functionality (tetraphenylol ethane glycidyl ether) [86], which enhanced ionic conductivity of 10 × 10^−3^ S cm^−1^ at 30 °C. Fang et al. reported a hydroxyl-conducting anion membrane prepared by introducing chloromethyl and quaternary ammonium groups into poly (phtalazinone ether sulfone) (PPESK), which showed good ionic conductivity [87]. Similarly, Gu et al. synthesized AEMs using tris (2,4,6-trimethoxyphenyl) phosphine with polysulfone, having quaternary phosphonium as the cationic head group [31].

General Electric was the first to develop AEMs based on polyphenylene oxide, with ammonium groups linked to PPO exhibiting an IEC of 3.80 meq. g^−1^ [88]. Hibbs et al. prepared AEMs based on poly (phenylene) backbone synthesized via a Diels–Alder reaction, followed by functionalization via bromomethylation and quaternization [89]. Xu et al. prepared chloroacetylated PPO and bromomethylated PPO separately, followed by blending the two polymers [90,91]. Finally, the membrane was quaternary-aminated using triethylamine. Katzfuß et al. utilized N-bromosuccinimide (NBS) as a brominating reagent to create an AEM from 1,5-dimethylpolyphenylenoxide. To generate a covalently cross-linked membrane, the brominated PPO (BrPPO) was further cross-linked using DABCO [92]. Wu et al. prepared AEMs by incorporating silica into PPO, which exhibited high tensile strength and good conductivity [93]. From the above discussion, it is clear than in relation to all the preparation methods mentioned, homogeneous membranes achieve enhanced APEFC performance in terms of conductivity and chemical stability due to the uniform distribution of fixed ionic charges over the entire polymer matrix.

## 6. Typical Numerical Model for AEMFC

A steady-state H_2_/O_2_ AEMFC numerical model was demonstrated by Peng et al. [94] to simulate the deviations in the polarization curve studies. This phenomenological model relates to the diffusion of reactants and products in the MPLs, catalyst layer, and diffusion layer, dependent on the concentration of Butler–Volmer kinetics in the catalyst and diffusion layers and the electro-osmotic drag of water molecules in the AEM. The transport parameters of the Tokuyama A201 membrane were subjected as a standard case for the AEM. The governing equations and boundary conditions were implemented and solved in a coupled manner in a 2D configuration using computational fluid dynamics. Typical simulated numerical cell polarization curves along with experimental validation for the various AEMs are represented in Figure 11.

Similarly, an isothermal, one-dimensional, steady-state model for an alkaline anion exchange membrane fuel cell was developed by Y J Sohn et al. [95]. The water transport in the membrane comprises water flux by the electro-osmotic drag and the diffusive water flux because of the gradient of water concentration across the membrane. The present model is expected to be useful for the proposed AEMFC systems. In the present model, the water exists only in vapor form. However, cell performance is affected by flooding at the anode under 100% humidity.

## 7. Mechanical and Transport Properties

The mechanical properties of membranes for fuel cell application are crucial, affecting their performance and durability, both in short- and long-term operation. In short-term operation, the water uptake is influenced by polymer relaxation, whereas in the long term, membrane creeps are formed due to stresses [96,97]. Any polymeric membrane material, depending on its viscoelasticity, responds to stress with relation to time, due to which the polymer flows, followed by its corresponding dynamic change in hydration [98]. The stresses can be induced in two ways: by clamping the polymer membrane during cell assembly or by changing the hydration levels [99]. As a result of these stresses, membrane thinning occurs, creating hot spots, often leading to pinhole formation, delamination between electrode and membrane, and insufficient contact between membrane and electrode, thereby reducing the performance [100]. Temperature and water content of the membrane strongly affect the viscoelastic behavior, affecting the operation and performance of fuel cells.

The mechanical integrity of membranes is subjected to various stresses, due to reasons such as cell assembly, disturbance of pressure gradients, swelling–dehydration cycles, mechanical creep, high temperatures, onset of brittleness, and tear resistance [101,102]. The long-term lifetime of a fuel cell stack is dependent on the membrane’s capability to resist mechanical degradation. In an ideal fuel cell operation, one can observe dynamic temperature and humidity conditions which result in a change in the membrane hydration level [103]. Water sorption and desorption can have an effect of significant swelling and hygrothermal stresses. These defects will deteriorate the membrane property, leading to mechanical failure and fuel crossover, which results in radical formation degrading the membrane chemically [104,105].

The polymer backbone chemistry accounts for the varying mechanical properties of different types of anion exchange membranes (AEMs). Based on research literature/reports, it has been inferred that an increase in polymer crystallinity can improve the mechanical properties by varying the polymer chemistry, i.e., by decreasing the side chain length, blending with an additive, or through optimizing the annealing techniques [106,107]. Chemical cross-linking is another technique to improve the membrane stability by cross-linking the polymer chains, thereby increasing the tensile modulus and strength. The limitation with this method is that cross-linking might reduce the ionic functional moieties, reducing the conductivity, and a higher degree of cross-linking causes membrane embrittlement [108]. Reinforcing an ion exchange membrane with nonconductive porous polymer fibers helps to mitigate dimensional swelling and increases the ability of withstanding dry and hydrated states, which prolongs the membrane lifetime [104]. Meanwhile, chemically stable and better performing AEMs are not available commercially and thus, various polymer chemistries are investigated; optimizing polymers based on mechanical properties will help to develop durable membranes. However, AEM development is still in its preliminary growth and a suitable mechanical testing method is not established worldwide; thus, comparing the mechanical properties of AEMs is difficult.

Recently, there has been immense research interest and an increase in publications on the development of AEMs. In spite of this, the overall membrane characteristics such as conductivity, performance, mechanical strength, and chemical stability are still inferior to those of PEMs. To overcome this, a rational approach in needed in materials design, which requires a thorough understanding of intrinsic micro-structures and the elementary ion transport mechanisms. Pan et al. exhibited preliminary modeling results on a quaternary ammonium polysulfone (QAPS) membrane which showed percolated ionic channels with distinct hydroxide ion distribution at various hydration levels [109]. Han et al. compared the results using polysulfone as backbone and explained the diffusion constant difference between AEM and PEM from the standpoint of ionic group solvation and free ion correlation [110]. Herbst et al. presented a theoretical study on the spatial distribution of water molecules absorbed on an AEM material. They postulate that when phase separation occurs between the hydrophilic moiety and water, the conductivity can be enhanced [111]. Tuckerman et al. [18] predicted the solvation structure of hydroxide in poly (vinyl benzyl trimethylammonium), wherein the OH^−^ forms hydrogen bonds with four adjoining water molecules while providing one bond to transient water positioned above the hydroxide hydrogen atom.

Hydration is an important process for producing continuous water-filled channels facilitating hydroxide transport by Grotthuss hopping, faster than that of diffusive transport. In water, both proton (H^+^) and hydroxide ions (OH^−^) are transported by structural diffusion (Grotthuss hopping) in addition to vehicular transport. In a hydrated environment, the solvation and transport of hydroxide are influenced by the Grotthuss mechanism [112], wherein the negative charge travels through a series of water molecules by means of O–H bond breaking and formation process. Chen et al. developed a multiscale and multicomplex model for OH^−^ transport, and found that phase difference of rigid side chains creates a continuous overlapping region [112]. The authors claim that hydroxide prefers this overlapping region, with transport through that region and between the polymer side chains with significant contributions from both vehicular (standard diffusion) and Grotthuss (proton hopping) mechanisms. The modular structure of fuel cells, along with their great efficiency, led to a wide range of applications in the stationary and automobile sector. As a consequence of advancements in the materials and design of fuel cell components in APEFC, several prototypes and scale demonstrations have already been developed. Figure 12 is block diagram of the direction towards the development of alkaline electrolyte membranes and future prospects.

## 8. Conclusions

The current review attempts to describe the significant research over the years to develop superior anion exchange membranes (AEMs) for use in fuel cell applications. Realizing the ion transport mechanism and significant efforts to boost conductivity while preserving stability are required before AEM can be utilized in APEFC. In contrast to the conventional approach of benzyl substitution, chloromethylation of polymer on the aryl carbon renders it less susceptible to crowding. Due to the ease of synthesis, the quaternary ammonium cation is the most extensively investigated anion conductive core class in AEMs so far. The effect of chemical modification by altering the amino group to polymers ratio was noteworthy since effective quaternization not only boosts ionic conductivity but it also retains membrane longevity. Anion membranes synthesized by quaternization of aliphatic polymers by adding organic salts containing quaternary ammonium group accompanied by ion exchange have been found to show promising performance. The progress is remarkable in terms of improving the anion conductivity; however, challenging factors in the APEFC, such as exposure of AEM to CO_2_ and SN_2_ nucleophilic attack, must be addressed in a systematic manner. A future research objective for AEMs would be to improve the stability and anion conductivity through various routes, such as introducing multi-cation groups for quaternization of polymers in addition to nanofibrous AEMs containing imidazolium-functionalized side chains. Studies in this direction are in progress.

## Figures and Tables

**Figure 1 materials-15-05601-f001:**
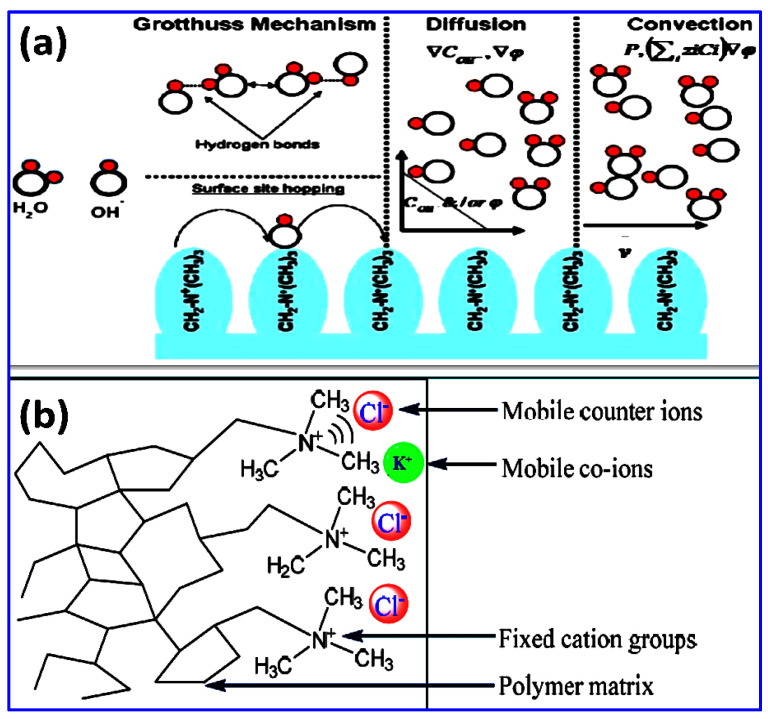
(**a**) Ion transport phenomenon of various mechanisms involving in anion exchange membrane; (**b**) interaction between polymer matrix and mobile counter ions and co-ions.

**Figure 2 materials-15-05601-f002:**
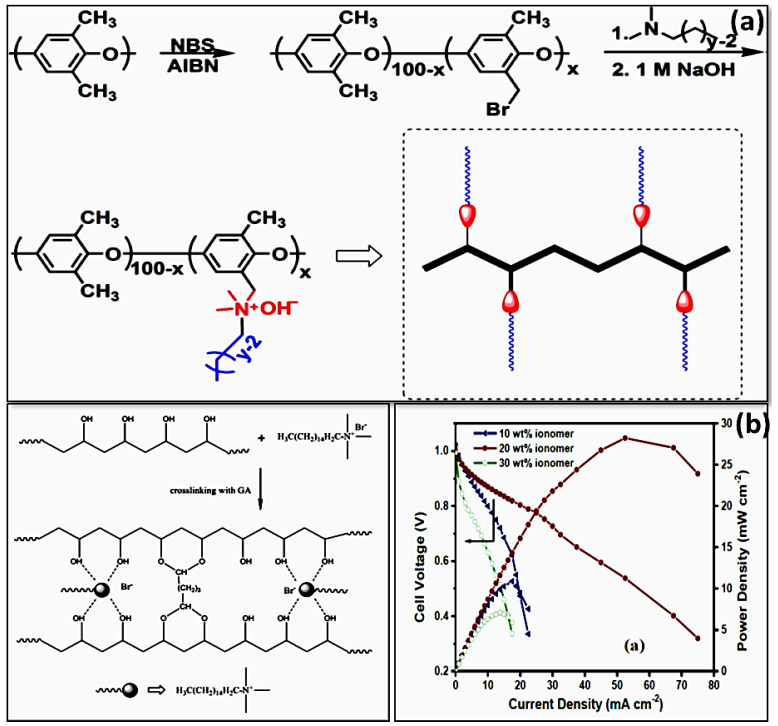
(**a**) Various steps in the synthesis of comb-shaped CyDx copolymers for APEFC application. (Reproduced with permission from Ref. [38] Copyright © 2013.) (**b**) Formation of quaternization of PVA by using hexadecyl trimethyl bromide (HDT) and its cell performance in relation to different wt % of composite ionomer. (Reproduced with permission from Ref. [41] Copyright © 2018).

**Figure 3 materials-15-05601-f003:**
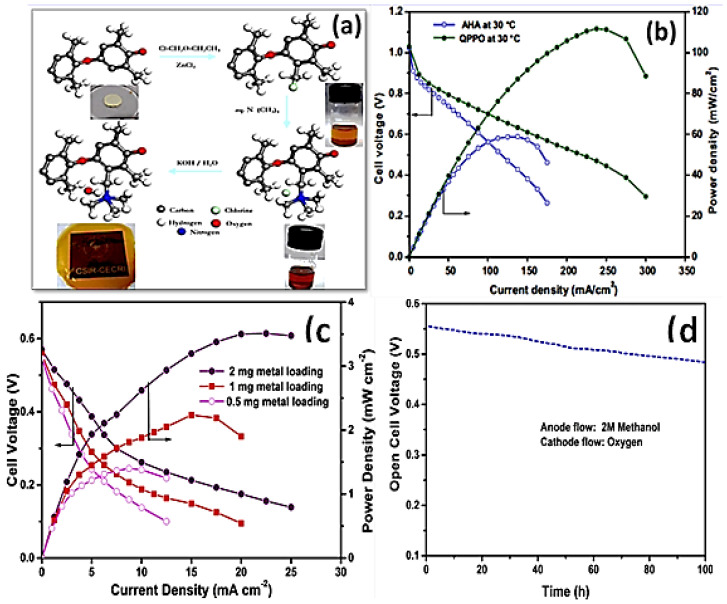
(**a**) Ball and stick model representing the preparation of quaternary ammonium-functionalized poly(2,6-dimethyl-1,4-phenylene oxide) as an anion exchange membrane (AEM). (Reproduced with permission from Ref. [42] Copyright © 2016.) (**b**) Cell polarization studies for AEM. (Reproduced with permission from Ref. [43] Copyright © 2014.) (**c**) ADMFC polarization of QPPO 3 membrane at 30 °C with different catalyst loading in KOH free fuel. (**d**) Long-time durability study using optimized QPPO 3 membrane at OCV condition. (Reproduced with permission from Ref. [42] Copyright © 2016).

**Figure 4 materials-15-05601-f004:**
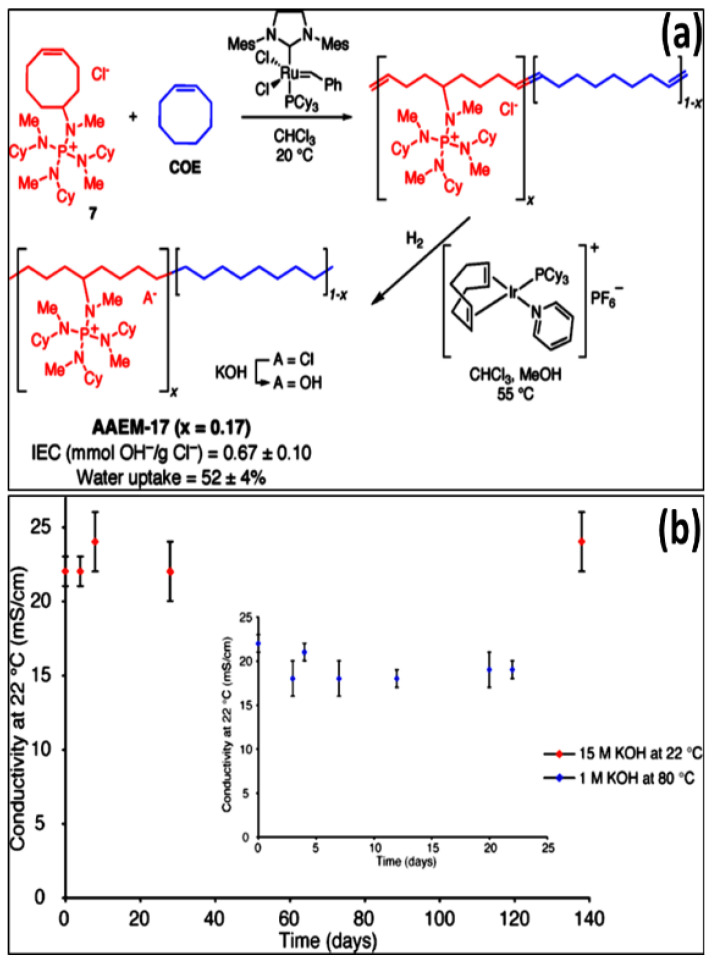
(**a**) Preparation Formation of phosphonium-functionalized polyethylene. (**b**) The conductivity of hydroxide as time-dependent following soaking in 15 M KOH (aq) at 22 °C. Inset: The conductivity of AAEM-17 hydroxide with respect to time following immersion in 1 M KOH (aq) at 80 °C. (Reproduced with permission from Ref. [46] Copyright © 2012).

**Figure 5 materials-15-05601-f005:**
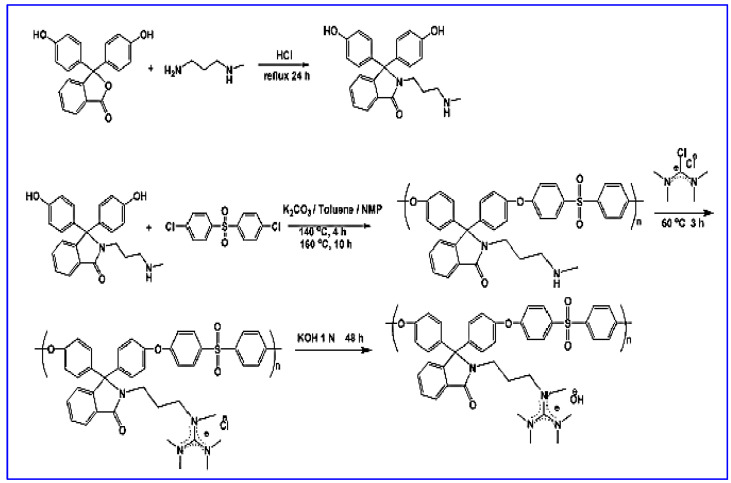
Preparation of poly(aryl ether sulfone) comprising hexa-alkylguanidinium framework (PES-G-OH). (Reproduced with permission from Ref. [29] Copyright © 2010).

**Figure 6 materials-15-05601-f006:**
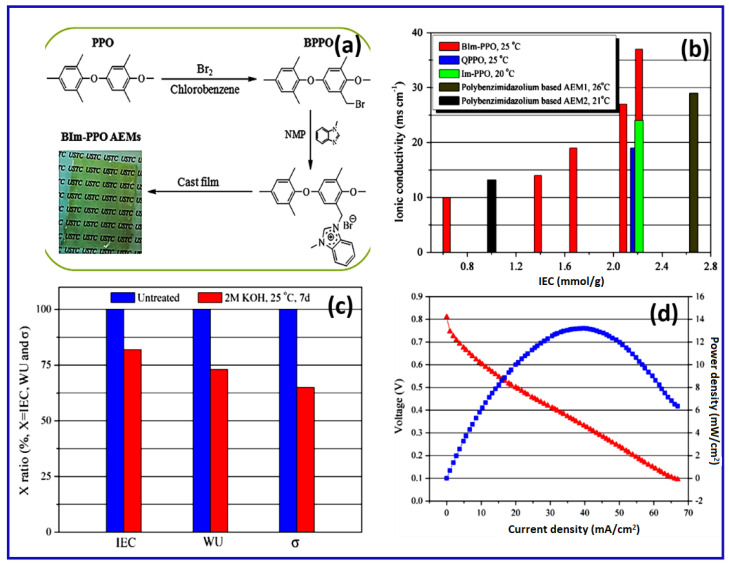
(**a**) Synthetic route of benzimidazolium-PPO AEMs for fuel cell application. (**b**) As a function of IEC, the ionic conductivity of Bim-PPO, the standard QPPO and Im-PPO, and polybenzimidazolium-based AEMs. (**c**) IEC, WU, and ionic conductivity % of BIm-PPO-0.54 AEM before and after immersion in an aqueous KOH solution (2 mol cm^−3^) at 25 °C for 7 days. (**d**) H_2_/O_2_ fuel cell performance curves at 35 °C with BIm-PPO-0.54 AAEM at start-up. (Reproduced with permission from Ref. [35] Copyright © 2013).

**Figure 7 materials-15-05601-f007:**
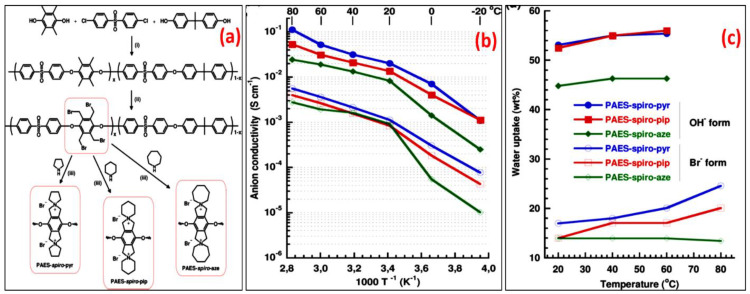
(**a**) Preparation pathway to PAES-spiro-pip, PAES-spiro-pyr, and PAES-spiro-aze via poly condensation, benzylic bromination, and cyclo-quaternization. (**b**) Arrhenius conductivity plots of fully hydrated (immersed) AEMs. (**c**) Water uptake data at different temperatures. (Reproduced with permission from Ref. [33] Copyright © 2015).

**Figure 8 materials-15-05601-f008:**
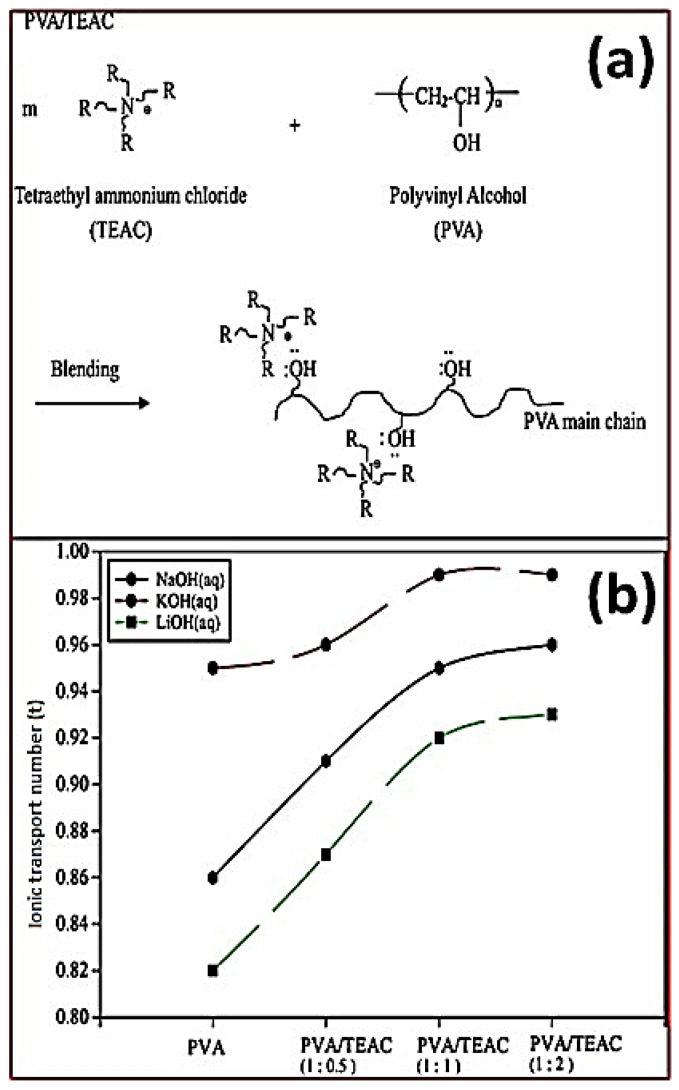
(**a**) PVA blends with tetraethyl ammonium chloride to form AEM. (**b**) The effect of alkali metal salts on anionic transport number for various PVA/TEAC blend conductive polymers. (Reproduced with permission from Ref. [58] Copyright © 2005).

**Figure 10 materials-15-05601-f010:**
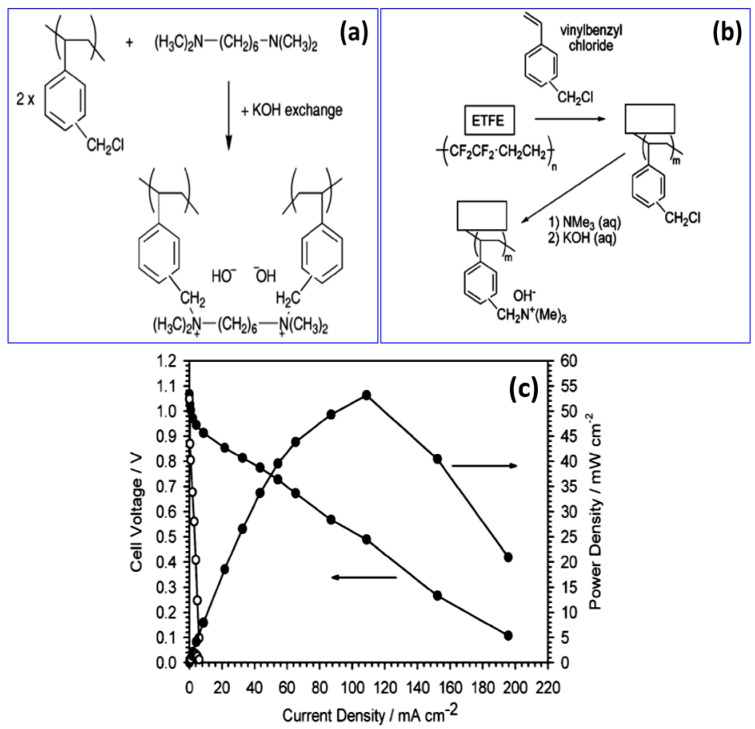
(**a**) Cross-linked alkaline interface polymer synthesis. AAEMs do not contain any counter ions (quaternary ammonium groups) to the OH^−^ conduction species. (Reproduced with permission from Ref. [72] Copyright © 2006.) (**b**) Synthesis of ETFE-based radiation-grafted alkaline anion-exchange membrane (ETFE-AAEM). (Reproduced with permission from Ref. [74] Copyright © 2007.) (**c**) I-V characteristics of AEM MEA Fuel cell test observed with AAEM-MEA with 0.5 mg/cm^2^ Pt/C (20 wt %) electrodes coated with the alkaline interface polymer (●), and with 4 mg/cm^2^ Pt black electrodes without the interface (O). (Reproduced with permission from Ref. [72] Copyright © 2006).

**Figure 11 materials-15-05601-f011:**
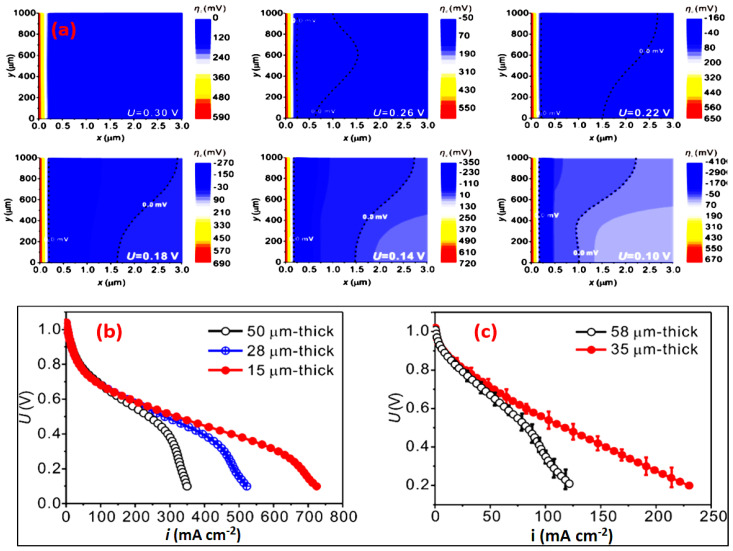
(**a**) High magnification for the cathode overpotential (ηc) distributions between x = 0 μm and 3 μm in the cathode catalyst layer. Range from U = 0.30 V to U = 0.10 V, where the drop segment and deviation segment were observed in the simulated polarization curve for the baseline case. The inert region (where ηc is below zero) is indicated with a dashed line. (**b**,**c**) Simulated polarization curves with different AEM thicknesses; experimentally measured polarization curves for 58 and 35 μm thick AEMs. (Reproduced with permission from Ref. [94] Copyright © 2015).

**Figure 12 materials-15-05601-f012:**
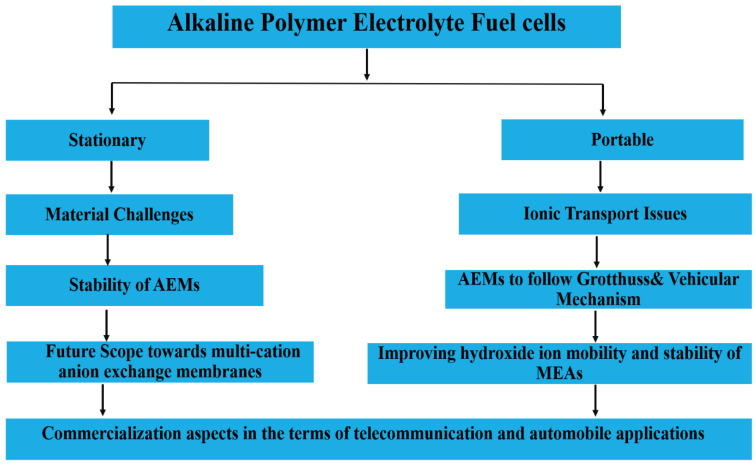
Flow chart for the future developments in AEMs.

**Table 1 materials-15-05601-t001:** Different cation head groups for ionic conductivity, IEC, and water uptake for AEM.

Sl. No.	Polymer	Cation Head Group	Ionic Conductivity mS/cm @ RT	IEC Meq/g	Water Uptake at RT	Ref.
1	Poly(2,6-dimethyl phenylene oxide) (PPO)	Quaternary ammonium (QA) type	10–28	1.10–2.08	10–59%	[38]
2	Poly(2,6-dimethyl phenylene oxide) (PPO)	Quaternary ammonium (QA) type	5–60 @ 60 °C	0.80–1.80	10–40%	[39]
3	Poly(2,6-dimethyl phenylene oxide) (PPO)	Quaternary ammonium (QA) type	5–35	1.08–1.92	8–26%	[40]
4	Poly(2,6-dimethyl phenylene oxide) (PPO)	Quaternary ammonium (QA) type	5–10	0.90–1.40	45–65%	[42]
5	Polysufone (PSf)	Quaternary phosphonium type	27	1.09		[31]
6	Polystyrene/Polysufone	Quaternary ammonium and phosphonium type		0.85–1.20	10–25%	[44]
7	Polysufone (PSf)	Quaternary phosphonium type	5–15	1.30–2.0	10–45%	[45]
8	Polyethylene	Quaternary phosphonium type	22	0.67	40–50%	[46]
9	Poly(aryl ether sulfone)	Guanidinium type	20–25	1.39	31%	[29]
10	Poly(aryl ether oxadiazole)s	Guanidinium type	10	0.86		[47]
11	Poly(2,6-dimethyl phenylene oxide) (PPO)	Imidazolium type	10–37	0.65–2.20	8–32%	[35]
12	Poly [2,2′-(*p*-oxydipheny lene)-5,5′-bibenz imidazole]	Benzimidazolium type	29			[48]

## Data Availability

Not applicable.

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
