# Peer review of "Anion Exchange Membranes for Alkaline Polymer Electrolyte Fuel Cells—A Concise Review"

_materials, 2022, doi:10.3390/ma15165601_

Round 1

Reviewer 1 Report

Recently, fuel-based energy systems, particularly alkaline polymer electrolyte fuel cells (APEFC), are being perceived as low-carbon energy technology that helps with energy security and sustainability while also reducing global warming and environmental effect. This review paper discussed several types of polymer-based or filler-supported membrane preparation and their fuel cell performance for APEFC applications. Especially, The Quaternary ammonium (QA), Phosphonium, Guanidinium, Benzimidazolium, Pyrrolidinium, and Spirocyclic cation-based AEMs are widely studied in the literature. Besides, the present review article gives a clear indication to understand the ionic conductivity enhancement and long-term durability. This revised article is interesting, and this paper can be published after the following minor revisions:

  1. The discussion part needs to be improved in Figure 1 based on the mechanism involved.
  2. Please check carefully this manuscript again and avoid some abbreviations.
  3. The overall grammar should be polished; there are some rough sections that are hard to read.
  4. How does a catalyst affect the overall?
  5. Some advanced and recent electrochemical papers are to be cited/ added.
  6. Why the Quaternary ammonium (QA) groups is essential for APEFC application?
  7. The author needs to discuss the durability part using Poly(phenylene oxide) (PPO) based membrane-based AEM for APEFC application.
  8. What is the ratio of Poly(ethylene oxide) (PEO) used in membrane to improve fuel cell performance?

Author Response

Reviewer #1: Recently, fuel-based energy systems, particularly alkaline polymer electrolyte fuel cells (APEFC), are being perceived as low-carbon energy technology that helps with energy security and sustainability while also reducing global warming and environmental effect. This review paper discussed several types of polymer-based or filler-supported membrane preparation and their fuel cell performance for APEFC applications. Especially, The Quaternary ammonium (QA), Phosphonium, Guanidinium, BenzimidazoliumPyrrolidinium, and Spirocyclic cation-based AEMs are widely studied in the literature. Besides, the present review article gives a clear indication to understand the ionic conductivity enhancement and long-term durability. This revised article is interesting, and this paper can be published after the following minor revisions:

General Response: The authors thank the reviewer for his/her valuable time spent reviewing the manuscript. As per the suggestions of the reviewers, the manuscript has been revised as appended below. Authors' response / Rebuttal for each of the query have been given below and the corresponding revisions if any are highlighted in yellow in the revised manuscript.

Comment 1: The discussion part needs to be improved in Figure 1 based on the mechanism involved.

Authors' Response: As per the suggestions of the reviewer, Figure 1 detailed mechanism involving in APEFC is now included in the revised manuscript in page 3/ paragraphs 3/ line 6-13.

Comment 2: The overall grammar should be polished; there are some rough sections that are hard to read.

Authors' Response: We have now carefully examined the grammar in the revised manuscript as per the suggestions of the reviewer. All the other minor errors are corrected now in the revised manuscript and highlighted in yellow.

Comment 3: How does a catalyst affect the overall performance?

 Authors' Response: Generally, during real time or stringent operating condition of APEFC application, first catalyst corrosion, followed by catalyst/membrane degradation and dissolution take place which results in decrease in APEFC performance gradually. Usually, APEFC catalyst loading is maintained between 0.3 to 1 mg /cm-2 for anode and cathode respectively. Moreover, with increased catalyst loading, the overall fuel cell performance increased. However, in the present review article our focus was mainly on the different type of AEM membranes to improve the overall fuel cell performance and durability in APEFC application. 

Comment 4 Some advanced and recent electrochemical papers are to be cited/ added.

Authors' Response: As per the suggestions of the reviewer, recent articles are now cited and included in the revised manuscript in page 2/ paragraphs 1/ line 1-7.

Comment 5 Why the Quaternary ammonium (QA) groups is essential for APEFC application?

Authors' Response: In general, quaternary ammonium (QA) groups provide comparatively high OH- conductivity, adequate alkaline stability throughout time scale, and simplicity of surface modification. Recent reports suggested that, the decomposition of the AEMs is dependent on the aliphatic chain length of amines, hence longer the alkyl chain length, lower the degradation of AEMs. QA groups that are attached or copolymerized with long alkyl N-bound sequences has strong alkaline durability. Besides, the steric effects of the long alkyl chains surrounding the QA group may improve the alkaline stability. The above discussion now included in the revised manuscript in page 5/ paragraphs 3/ line 1-8.

Comment 6 The author needs to discuss the durability part using Poly(phenylene oxide) (PPO) based membrane-based AEM for APEFC application.

Authors' Response:  As per the suggestions of the reviewer, the sentences pertaining to durability part using optimum level of poly(phenylene oxide) (PPO) based composite membrane is now included/discussed in the revised manuscript in page 8/ paragraphs 1/ line 10-14.

Comment 7: What is the ratio of poly(ethylene oxide) (PEO) used in membrane to improve fuel cell performance?

Authors' Response: As per the suggestions of the reviewer, the various ratio of poly(ethylene oxide) composite membrane and its effect of fuel cell performance are now included in the revised manuscript in page 14/ paragraphs 2/ line 9-14.  Once again authors thank the reviewer for his/her valuable time spent reviewing the manuscript.

Reviewer 2 Report

please find comments in the attachment.

Author Response

Reviewer #2:

Comment 1: The image format in the paper is inconsistent Indicate the source of the image and reference danger; add it to the figure title.

Authors' Response: As per the suggestions of the reviewer, the relevant figures, figure captions and references are now revised in the manuscript. 

Comment 2: The paper has used a large number of references. But there are not many corresponding pictures, only 10, The author should add more reference images,

Authors' Response: As per the suggestions of the reviewer, the relevant figures are now included in the revised manuscript. 

Comment 3: The corresponding mechanism is not explained much, especially the mechanical mechanism of particle motion needs more explanation. A chapter can be added to explain the direction of movement of particles inside the relevant battery, the speed of movement, and the elaboration of the life problem caused by the blockage of moving particles [1-3].

Authors' Response: Authors thank the reviewer for the important suggestion. Authors would wish to clarify here the focus of the review article is the feasibility of AEM in APEFC application. Discussion pertaining to ionic transport through the membranes is now expanded in the revised manuscript. 

Comment 4: Can the author give a block diagram of the development direction of fuel cells

Authors' Response: Authors thank the reviewer for the important suggestion. As per the suggestions of the reviewer, a block schematic on the development direction of APEFC is included in the revised manuscript on page 23/ paragraphs 1/ line 1-5 and Figure 12.

Comment 5: The overall feel of the paper: The author still needs to add much content, especially the scientific content.

Authors' Response: The current review article has been modified accordingly with more details and the explanations are marked in yellow in the revised manuscript.

Comment 6: Numerical calculation of fuel cells is very important. It is suggested that the author add this section and add more test references and pictures.

Authors' Response: Authors thank the reviewer for the important suggestion. Important literature and explanation pertaining to numerical model in APEFC is now included in the revised manuscript on page 21/ paragraphs 3/ line 1-9 and page 22/ paragraphs 1/ line 1-8 and Figure 11.

Round 2

Reviewer 2 Report

 Author has not cite references suggested.

Author Response

Reviewer #2:

Comment 1: Author has not cite references suggested.

Authors' Response: Authors thank the reviewer for the important suggestion. The relevant mechanical, and transport properties are now included in the revised manuscript on page from 24 to 26. Reviewer suggested relevant references on fuel cells are now included in the revised manuscript.
